# A Newly Defined CHA_2_DS_2_-VA Score for Predicting Obstructive Coronary Artery Disease in Patients with Atrial Fibrillation—A Cross-Sectional Study of Older Persons Referred for Elective Coronary Angiography

**DOI:** 10.3390/jcm11123462

**Published:** 2022-06-16

**Authors:** Zyta Beata Wojszel, Łukasz Kuźma, Ewelina Rogalska, Anna Kurasz, Sławomir Dobrzycki, Bożena Sobkowicz, Anna Tomaszuk-Kazberuk

**Affiliations:** 1Department of Geriatrics, Medical University of Bialystok, Fabryczna Str. 27, 15-471 Bialystok, Poland; 2Department of Invasive Cardiology, Medical University of Bialystok, M.C. Skłodowskiej Str. 24a, 15-276 Bialystok, Poland; kuzma.lukasz@gmail.com (Ł.K.); annaxkurasz@gmail.com (A.K.); slawomir.dobrzycki@umb.edu.pl (S.D.); 3Department of Cardiology, Medical University of Bialystok, M.C. Skłodowskiej Str. 24a, 15-276 Bialystok, Poland; ewelina.rogalska@umb.edu.pl (E.R.); sobkowic@wp.pl (B.S.); a.tomaszuk@poczta.fm (A.T.-K.)

**Keywords:** obstructive coronary lesions, geriatric patients, predictive factors, chronic coronary artery disease

## Abstract

Purpose: Atrial fibrillation (AF) can be a valuable indicator of non-obstructive coronary artery disease (CAD) among older patients indicated for elective coronary angiography (CAG). Appropriate stratification of AF patients is crucial for avoiding unnecessary complications. The objective of this study was to identify independent predictors that can allow diagnosing obstructive CAD in AF patients over 65 years who were indicated to undergo elective CAG. Patients and methods: This cross-sectional study included 452 (23.9%) AF patients over 65 years old who were directed to the Department of Invasive Cardiology at the Medical University of Bialystok for elective CAG during 2014–2016. The participants had CAD and were receiving optimal therapy (median age: 73 years, interquartile range: 69–77 years; 54.6% men). The prevalence and health correlates of obstructive CAD were determined, and a multivariate logistic regression model was generated with predictors (*p* < 0.1). Predictive performance was analyzed using a receiver-operating characteristic (ROC) curve analysis. Results: Stenosis (affecting ≥ 50% of the diameter of the left coronary artery stem or ≥70% of that of the other important epicardial vessels) was significant in 184 (40.7%) cases. Multivariate regression analysis revealed that only the male sex (odds ratio [OR]: 1.80, 95% confidence interval [CI]: 1.14–2.84, *p* = 0.01) and the newly created CHA_2_DS_2_-VA score (OR: 3.96, 95% CI: 2.96–5.31, *p* < 0.001) significantly increased the chance of obstructive CAD, while controlling for chronic kidney disease and anemia. The ROC curve analysis indicated that the CHA_2_DS_2_-VA scale may be a useful screening tool for the diagnosis of obstructive CAD (area under the ROC curve: 0.79, 95% CI: 0.75–0.84, *p* < 0.001), with ≥4 being the optimal cutoff value. Conclusions: Our study has proven that several older AF patients who are advised to undergo elective CAG have nonobstructive CAD. The CHA_2_DS_2_-VA score can contribute to improving the selection of patients for invasive diagnosis of CAD, but further investigation is required.

## 1. Summary

As elective coronary angiography (CAG) generally reveals no significant coronary lesions, it is important to improve patient stratification. This is particularly true in the case of older adults and persons with atrial fibrillation (AF), which reflects the difficulties in qualifying patients for CAG. Thus, it is essential to explore this topic in the AF population.

A cross-sectional, single-center study was carried out on 452 AF patients aged over 65 years, who were indicated for elective CAG between 2014 and 2016 in Poland. More than half (59.3%) of elective cardiac catheterizations revealed no significant coronary artery plaques. Male sex and a high CHA_2_DS_2_-VA score were associated with a higher diagnostic yield of elective CAG in the studied AF population.

In conclusion, the CHA_2_DS_2_-VA score can improve the qualification of AF patients for invasive diagnosis of CAD. It allows distinguishing individuals who have and do not have obstructive CAD, with ≥4 being the optimal cutoff value.

## 2. Introduction

By 2050, the global proportion of persons who are ≥65 years old is projected to reach nearly 16%, and by 2100 to nearly 23%. The share of people over 80 years, which in 2019 was estimated at 143 million, is growing even faster and is projected to triple by 2050 and increase further to 881 million in 2100 [1]. The prevalence of diabetes in industrialized nations is also increasing due to population aging and the expanding obesity epidemic [2,3]. Although the proportion of diabetes cases worldwide was 9.3% in 2019 (463 million), it will increase up to 10.9% by the year 2045 (700 million) [4]. Older age, obesity, diabetes, and coexisting hypertension or hyperlipidemia are the main risk factors of CAD [5]. Their high prevalence and increasingly improving therapeutic methods will increase the necessity for therapeutic decisions regarding strategies for treating acute coronary syndrome (ACS) and stable coronary artery disease (CAD) in older populations [6,7].

Early invasive treatment and revascularization is beneficial in frail older adults with ACS [8,9]. Although previous studies have not proven that invasive treatment of stable angina, an important clinical symptom of CAD observed among older adults, will always result in lower mortality rates compared to conservative treatment, recent research suggests that percutaneous coronary angioplasty (PTCA) using new-generation drug-eluting stents and coronary artery bypass grafting may not only improve the well-being of older patients affected by chronic CAD but also positively influence the prognosis [10]. From the healthcare organizers’ point of view, cost-effectiveness analysis results are also critical, as they indicate that PTCA treatment in older patients can cut treatment costs in the long run [11]. Therefore, it may be expected that in the coming years there will be an increase in cardiac catheterization in elderly patients to assess the need for surgical or PTCA treatment of stable angina.

As coronarography can pose more risks in the elderly [12,13,14,15,16,17,18], especially in frail older patients [19] precise selection of patients for this procedure is important (eliminating the risk of overtreatment). The procedure should only be performed when there is a high probability of significant coronary stenosis.

Several studies on elective coronary angiography (CAG) have confirmed that in a large percentage of patients cardiac catheterization does not confirm notable changes in coronary vessels, and that the patients remain on conservative therapy. In a US study using the College of Cardiology National Cardiovascular Data Registry, significant stenosis was observed in 62.4% of relatively young patients undergoing elective CAG between 2004 and 2008 in 663 hospitals [20]. This result suggests older age to be an independent risk determinant. Our previous study in older patients (65+ years of age) showed that the prevalence of nonobstructive CAD is high and that stratification of patients for invasive CAD diagnosis, including those with AF, should be improved [21]. As more sensitive and specific stratification strategies may enable better utilization of health care resources for diagnostic applications, the objective of the study was to find independent determinants of obstructive CAD among older AF patients who are indicated for elective CAG. For instance, it has been found that a combined assessment of risk factors for ischemic heart disease, as well as indicative of a risk of thromboembolism in AF, can help in evaluating the severity of CAD [22,23].

## 3. Materials and Methods

### 3.1. Participants and Study Design

This cross-sectional, single-center study included 1895 patients aged ≥ 65 (median age: 72 years, interquartile range [IQR]: 68–76 years; 50.3% women) who were directed to the Department of Invasive Cardiology at the Medical University of Bialystok to undergo elective CAG during 2014–2016. The final sample included a total of 452 (23.9%) cases with AF (median age: 73 years, IQR: 69–77 years; 54.6% men). This is a subanalysis of broader multicenter observational trial results devoted to AF and nonobstructive coronary lesions [24]. The study followed the 1975 Declaration of Helsinki guidelines and was registered in the clinical studies database (trial no. NCT04537507). Approval was obtained from the Medical University of Bialystok Ethical Committee (no. R-1-002/18/2019).

Consecutive older patients with AF who were advised for CAG because of worsened angina (with symptoms such as recurring or long-term chest discomfort/angina, typical angina, or others, such as dyspnea), despite being on optimal therapy for CAD according to the current standards, were included in the study. ACS, a previous diagnosis of ischemia or a moderate or severe disease of coronary valves, qualification for surgical replacement of heart valve, and Takotsubo cardiomyopathy were exclusion criteria (Figure 1). All patients received treatment based on the ESC Guidelines on Atrial Fibrillation and Chronic Coronary Syndrome during hospitalization, but precise information on the drugs prescribed was not available.

### 3.2. Study Parameters 

Data on variables characterizing patients were collected from hospital records. Patients’ age and sex, the prevalence of conditions that can elevate the risk of coronary heart disease (including AF, congestive heart failure, diabetes, high blood pressure, hepatic failure, hyperlipidemia, anemia), and coronary angiogram findings were noted. The diagnosis of the chronic coronary syndrome and the need for percutaneous coronary intervention were established based on the current European standards [25]. Stenosis affecting ≥ 50% of the left coronary stem diameter or ≥70% of the diameter of other primary epicardial vessels (“stenosis+” cases) was considered to indicate significant stenosis of the coronary vessels. Patients who did not meet this criterion were classified as “stenosis−”.

Assessment of left ventricular ejection fraction (LVEF) was carried out by transthoracic echocardiography following the modified biplane method of Simpson, in accordance with the guidelines of the European Society of Echocardiography [26]. AF (paroxysmal or chronic) was diagnosed by evaluating clinical records, monitoring for 24 h, and performing echocardiography (ECG) during admission [27].

Chronic kidney disease (CKD) was diagnosed following the KDIGO 2012 Clinical Practice Guideline for the Evaluation and Management of Chronic Kidney Disease [28]. An estimated glomerular filtration rate (determined using the CKD-EPI formula) of <60 mL/min/1.73 m^2^ for a minimum of 3 months indicated CKD. Body mass index was calculated for all patients, and obesity was confirmed if BMI was ≥30 kg/m^2^. A history of diabetes or the use of antidiabetic treatment indicated diabetes. Hypertension was diagnosed if systolic blood pressure was ≥140 mmHg, diastolic blood pressure was ≥90 mmHg, or the patient used antihypertensive medications, while hyperlipidemia was confirmed from a prior diagnosis or if the patient used antihyperlipidemic drugs. Anemia was confirmed if the hemoglobin level (estimated during admission) was <12 g/dL (7.45 mmol/L) in women and <13 g/dL (8.07 mmol/L) in the case of men. Hepatic failure was confirmed based on cirrhosis or if the level of bilirubin was ≥2 times the upper limit of normal (ULN), or if the level of transaminases or alkaline phosphatase was ≥3 times the ULN. The CHA2DS2-VASc score was calculated based on the following: C, congestive heart failure (or left ventricular systolic dysfunction)—1 point; H, hypertension—1 point; A2, age 75+ years—2 points; D, diabetes mellitus—1 point; S2, prior stroke, transient ischemic attack, or thromboembolism—2 points; V, vascular disease (e.g., peripheral artery disease, myocardial infarction, aortic plaque)—1 point; A, age 65–74 years; Sc, sex category (i.e., female sex—1 point) [29]. The maximum possible score on the scale was 9 points. The CHA2DS2-VASc scale is gender-sensitive and assigns 1 point for the female sex. As the male sex may be more associated with coronary heart disease risk, a nongender CHA2DS2-VA scale with a maximum score of 8 points was developed. We also assessed the baseline scale for assessing the risk of thromboembolic complications. The CHADS2 score was calculated based on the following: C, congestive heart failure (or left ventricular systolic dysfunction)—1 point; H, hypertension—1 point; A, age 75+ years—1 points; D, diabetes mellitus—1 point; S2, prior stroke, transient ischemic attack, or thromboembolism—2 points. The maximum possible score on the scale was 7 points [30].

### 3.3. Statistical analysis 

Data collection and analysis were carried out using IBM SPSS (v. 18, SPSS, Chicago, IL, USA). Variables distribution was assessed using the Kolmogorov–Smirnov test. For continuous variables with nonnormal distribution, the results are shown as median and IQR, and for categorical variables as the number of cases and percentage. Statistical significance of changes in the studied variables between the (“stenosis+” and “stenosis−”) groups was determined using the Mann–Whitney *U* and *χ*^2^ test. Odds ratios (ORs) were estimated to identify the risk determinants that can possibly have an impact on the frequency of obstructive CAD in older AF patients. In addition, multivariate logistic regression analysis with all the predictors (with *p* < 0.1) and lacking significant multicollinearity was performed. Correlations between variables, as well as their significance, were determined based on the variance inflation factor. Furthermore, a receiver-operating characteristic (ROC) curve analysis was carried out to assess the variables’ predictive performance. The optimal cutoff point that ensures high sensitivity and specificity was estimated with a confidence interval (CI). A two-tailed *p*-value of <0.05 was considered significant. In case of missing data, statistics were measured for appropriately reduced groups.

## 4. Results

Significant stenosis in coronary vessels was diagnosed only in 40.7% of cases. Table 1 provides the details of the stenosis+ and stenosis− groups. The median patient age was 73 years (IQR: 69–77 years) and was similar in both studied groups. The stenosis+ group had a higher proportion of men than the control group (65.2% vs. 47.4%, *p* < 0.001). No significant difference in the frequency of hypertension, obesity, diabetes, hyperlipidemia, congestive heart failure, hepatic failure, or in AF type was observed between the groups. The median values of BMI, eGFR and LVEF determined in the stenosis+ group were similar to that of the control group. Stenosis+ patients had significantly increased prevalence of CKD (45.7% vs. 36.2% in the stenosis− group, *p* = 0.04), vascular disease (89.1% vs.11.6%, *p* < 0.001) and significantly higher median CHA_2_DS_2_-VASc score (median: 4, IQR: 4–5 vs. median: 3, IQR: 3–4). Additionally, this group was characterized by a higher frequency of anemia (19.6% vs. 13.4%, *p* = 0.09), but significantly lower prevalence of stroke, TIA, or thromboembolism (13.0% vs. 13.0%, *p* < 0.001)

The CHA2DS2-VASc scale is female-based, and the risk of CAD is thus more remarkable for men. Regression analysis showed that the use of this scale resulted in a “strange” increase in OR for the male sex (data not presented). Therefore, it was decided that gender should be removed from the scale included in the regression analysis, since gender was considered separately as an independent variable. As a result, a genderless CHA2DS2-VA scale that more strongly correlated with obstructive CAD than the original CHA2DS2-VASc scale was developed, in which the OR for having obstructive CAD was nearly two times higher than in the case of the original scale. The newly created score did not cause “bizarre” changes in male OR in the regression analysis. 

Logistic regression was carried out considering “stenosis+” (outcome) and four important variables (namely sex (male), CKD, CHA_2_DS_2_-VA score, and anemia) (Table 2, Model 1). A significantly increased risk of obstructive CAD was found for the male gender (OR: 1.80, 95% CI: 1.14–2.84, *p* = 0.01) and CHA_2_DS_2_-VA score (OR: 3.96, 95% CI: 2.96–5.31, *p* < 0.001), when controlling for CKD and anemia. The rate of prediction success was determined at 75.0%, with an accurate prediction of 66.3% of “stenosis+” (sensitivity) and 81.0% of “stenosis−” (specificity) status.

The predictive performance of variables, namely CHA_2_DS_2_-VA scale score, male sex, CKD, and anemia, to differentiate between patients having and not having obstructive CAD was tested using the ROC curve analysis (Table 3). The test revealed that the CHA_2_DS_2_-VA score can serve as a predictor of significant stenosis in CAG. The area of the ROC curve was calculated at 79% for the CHA_2_DS_2_-VA score. The optimal cutoff value for this score was found to be 4.0 (area under the ROC curve [AUC]: 0.74, 95% CI: 0.69–0.79, *p* < 0.001), which yielded the best sensitivity (81.0%) and specificity (66.3%) for the prediction of obstructive CAD in CAG. This was confirmed by OR for this value of the scale (Table 1). 

AUC was also significant for the male sex, but it was significantly lower than that for CHA_2_DS_2_-VA score (Figure 2). 

To test the significance of CHA_2_DS_2_-VA score ≥ 4, Model 2, including sex (male), CKD, and anemia, was constructed. In the regression analysis, the CHA_2_DS_2_-VASc score ≥ 4 increased the risk of significant coronary stenosis (Table 2, Model 2). The rate of prediction success was determined at 75.0%, which was close to that of Model 1, with an accurate prediction of 66.3% of “stenosis+” (sensitivity) and 81.1% of “stenosis−” (specificity) status.

## 5. Discussion

Our study showed that in the group of older adults with AF undergoing elective diagnostic CAG, the proportion with clinically significant atherosclerotic lesions was only 40.7%. Although it is sometimes essential to have a “negative diagnosis”, older patients may be needlessly exposed to the risk of invasive diagnostic procedures in a majority of cases. Our results are consistent with earlier observations that as many as 62.4% of cardiac catheterizations in patients with CAD do not confirm the presence of significant changes in the coronary vessels [20]. Thus, it can be perceived as unnecessarily exposing patients to procedure-related risks and incurring unwanted health care costs [11].

In our study, AF was found in 23.9% of patients aged 65+ years who were indicated for elective CAG, which represents more frequency than general geriatric population. The high prevalence of identified risk determinants for both AF and CAD in the studied population (old age, male gender, hypertension, hyperlipidemia, obesity, CKD, diabetes, vascular disease, and heart failure) contributed to this phenomenon [31,32,33,34,35]. AF is one of the most prevalent arrhythmias among geriatric patients with CAD. In general, population aging, better survival rate of numerous diseases, and advances in diagnostics have led to a higher incidence of AF in developed countries [36].

Certain signs of acute symptomatic AF and that of CAD (dyspnea, ST depression at the time of a rapid fibrillation episode or elevated cardiac troponins—especially in patients with heart failure) often overlap. As a result, several patients are advised to test for CAD [37,38,39,40,41,42,43]. Patients with AF are a particularly problematic group for noninvasive testing, which at present is broad. The current guidelines of the European Society of Cardiology (ESC) on the diagnostic process of CAD place great emphasis on assessing the probability of the disease before the test (PTP) based on age, gender, and clinical symptoms—depending on the PTP score, additional imaging is recommended or further diagnosis is abandoned. Noninvasive methods of diagnosis include a stress test, rest and stress ECG, SPECT (single-photon emission tomography), PET (positron emission tomography), magnetic resonance, and cardiac computed tomography [25,44]. The limited physical capacity or frailty syndrome limits the possibility of performing stress tests in patients with AF, and the stress test results are mostly inconclusive [37,45,46,47,48]. These contribute to referrals for invasive diagnostics, exposing patients to potential risks [49]. 

The number of coronary interventions performed in elderly patients with chronic coronary disease, including elective ones, has increased worldwide as the prevalence of stable angina, defined according to the Canadian Cardiovascular Society classification III, increases with older age. The primary motivation to undergo these procedures in older patients is the desire to remain independent and mobile or to prevent a heart attack [50]. As research shows, the success rate of elective PTCA is relatively high, even in the elderly [51]. In addition, the risks related to these procedures are not much higher in older patients compared to younger adults. This can be attributed to the progress and improvement of the safety of intervention techniques [52]. Nevertheless, specific risks and difficulties are associated with CAG and PTCA in the elderly due to more complex morphology of the coronary vessels, frequency of multivessel disease, calcifications, or comorbidity, which lead to complications in pharmacotherapy, as well as polytherapy [53,54,55]. Thus, clinicians face a significant challenge in deciding whether to refer older patients with AF for elective CAG, and this is a multifactorial issue [56].

Various other factors may be responsible for the low obstructive CAD frequency observed in the present work. In previous studies, factors namely young age, female gender, uncommon presentation, lower complications in noninvasive diagnostic findings, low comorbidities, and absence of primary risk determinants of CAD were linked with the occurrence of nonobstructive CAD [20,57]. Suspected vascular spasm or microvascular disease in the studied group of patients may be the reason for their referral to CAG, even though the physicians do not expect to find significant coronary lesions. Moreover, in some cases, CAG is mainly planned to exclude significant stenosis in the coronary vessels rather than confirming its presence [58]. 

Although both stroke and coronary artery disease share common risk factors, we observed that obstructive coronary artery disease was significantly less frequently observed in patients with a history of stroke, TIA, or thrombotic events. In the recent study on the prevalence of CAD in ischemic stroke patients, when myocardial Stress–Rest Gated Technetium-99m (Tc99m) MIBI Myocardial Perfusion SPECT scan was performed on a dual-head SPECT-CT to estimate evidence of myocardial ischemia it was proved in 17.67% of cases, but no definitive relationship was found between coronary artery disease and intracranial or extracranial large artery cerebrovascular disease [59]. Perhaps the inverse relationship we observed results from changes in lifestyle or different therapies in patients with a history of cerebral incidents, but the data we collected did not allow us to verify this hypothesis. Our study did not prove that age had an influence on the prediction of heart catheterization findings in the studied older AF patients. Logistic regression showed that males had a >1.8-fold greater chance of having significant stenosis in coronary vessels and the probability increased significantly with a higher CHA_2_DS_2_-VA score. The chance of a positive elective CAG result was the highest if this score was ≥4. Although other studies have confirmed that CKD promotes the development of CAD [60], its relationship with anemia is unclear. CKD may cause hyperkinetic circulation, contributing to the damage to the vessel wall [61]. On the other hand, it may lead to false-positive results in noninvasive tests performed for the diagnosis of CAD [62].

The CHA_2_DS_2_-VASc scale is a well-known tool applied to evaluate the chances of thromboembolism in the AF population. According to recent studies, this tool can help in the prognosis of stroke in patients who do not have AF [63] and disorders of the heart and brain vasculature in patients having CAD without AF [64], as well as in the screening of AF [65]. Thus, an assessment of the degree of coexistence of risk factors assessed in the scale can allow making a well-targeted decision about whether or not to perform an invasive examination to determine the state of coronary vessels in CAD patients, which confirms earlier observations that the CHA_2_DS_2_-VASc score is a predictor of the severity of ischemic disease, especially when points for hyperlipidemia (H) and smoking (S) [22]—or additionally family history of premature CAD (F) [23]—are added. Unfortunately, due to the lack of information on smoking and the family history of CAD in the available database, a CHA_2_DS_2_-VASc-HS or CHA_2_DS_2_-VASc-HSF score could not be created. However, an attempt to add the hyperlipidemia variable had no significant influence on the results of the present analysis (data not shown). Additionally, contrary to other authors’ findings [23], in the case of the CHADS2 scale included in the analyzes, we did not observe significant differences in the incidence of obstructive CAD, which could result from double scoring of the stroke, TIA, or thromboembolic events in this scale (significantly more often reported by Stenosis-persons). Moreover, the newly developed non-gender CHA_2_DS_2_-VA scale correlated better with the odds of obstructive CAD than the classic CHA_2_DS_2_-VASc scale.

Our study was carried out on a large sample, depicting the actual clinical practice scenario of a well-established invasive cardiology institution, which is a strength. Nevertheless, the study has several limitations, such as a retrospective design, restricting the possibility of acquiring some data (use of natriuretic peptides, BMI, smoking); and the influence of visual evaluation of the angiographer on the judgment of stenosis severity which may lead to a higher margin of error [66]. In addition, the effect of ischemia related to coronary microvascular dysfunction was not assessed. The participants were not unsystematically recruited from different centers, but chosen from only one center, increasing the risk of selection bias. Therefore, it is important to carefully interpret the findings because the predictive performance of determining factors, such as the CHA_2_DS_2_-VASc or CHA_2_DS_2_-VA score, may vary for the general CAD community. Furthermore, AF was diagnosed primarily by ECG, and also 24 h monitoring. As a result, cases of “silent” AF in patients who were in the hospital may have been unidentified. The use of patients’ clinical charts for the detection of AF comorbid conditions may also cause some cases to be missed. 

## 6. Conclusions

Our study demonstrated that nonobstructive CAD may be frequent in older AF patients who are advised to go undergo elective CAG, due to difficulties in qualifying them for this procedure. Male sex and higher CHA_2_DS_2_-VA score were identified as the primary determinants increasing the probability of undergoing necessary invasive diagnostics, such as elective CAG, increasing its diagnostic yield in geriatric AF patients. A CHA_2_DS_2_-VA score of ≥4 may be a useful predictor for selecting patients for invasive CAD diagnosis, but it needs further research. CHA_2_DS_2_-VA can help in distinguishing individuals who have and do not have AF, with ≥4 being the optimal cutoff value, and CHA_2_DS_2_-VA assessment may be applied as the initial step in the diagnosis.

## Figures and Tables

**Figure 1 jcm-11-03462-f001:**
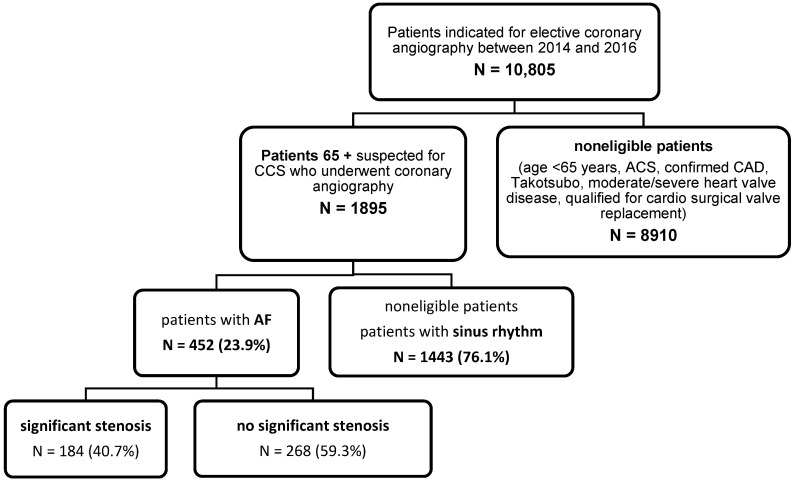
Flowchart of patients’ enrollment. Abbreviations: ACS, acute coronary syndrome; AF, atrial fibrillation; CAD, coronary artery disease; CCS, chronic coronary syndrome.

**Figure 2 jcm-11-03462-f002:**
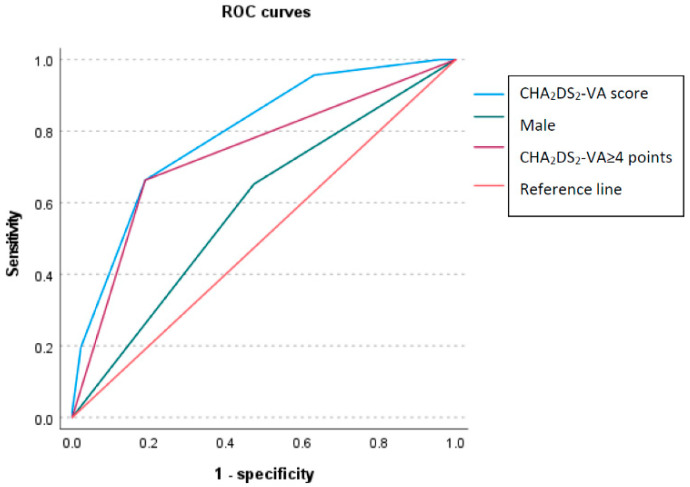
Predictive performance of CHA_2_DS_2_-VA score, CHA_2_DS_2_-VA score ≥ 4, and male sex—receiver-operating characteristic (ROC) curve analysis.

**Table 1 jcm-11-03462-t001:** Characteristics of the studied groups (N (%) or Me [IQR]).

	Total	Stenosis+	Stenosis−	*p*-Value	OR (95% CI)
N [%]	452 (100.0)	184 (40.7)	268 (59.3)		
Age, years	73.0 [69.0–77.0]	74.0 [68.0–77.0]	73.0 [69.0–76.0]	0.55	1.02 (0.98–1.05)
Sex, male	247 (54.6)	120 (65.2)	127 (47.4)	<0.001	2.08 (1.41–3.06)
BMI, kg/m^2^	29 [26.4–33.1] (n = 388)	29 [26.3–31.9] (n = 154)	29 [26.6–33.3] (n = 234)	0.33	0.98 (0.94–1.03)
Obesity	161 (41.5)	57 (37.0)	104 (44.4)	0.15	0.74 (0.48–1.11)
Hypertension	388 (85.8)	159 (86.4)	229 (85.4)	0.89	1.08 (0.63–1.86)
Diabetes mellitus	114 (25.2)	47 (25.5)	67 (25.0)	0.91	1.03 (0.67–1.59)
Hyperlipidemia	223 (49.3)	99 (53.8)	124 (46.3)	0.13	1.35 (0.93–1.97)
Chronic heart failure	143 (31.6)	51 (27.7)	92 (34.3)	0.15	0.73 (0.49–1.11)
Prior stroke, TIA, or thromboembolism	96 (21.3)	24 (13.0)	72 (27.0)	<0.001	0.64 (0.50–0.82)
Vascular disease	195 (42.3)	164 (89.1)	31 (11.6)	<0.001	62.4 (34.4–113.3)
LVEF, %	48 [35.0–56.5] (n = 225)	47.5 [36.5–57] (n = 84)	48 [32.5–56] (n = 141)	0.65	1.01 (0.99–1.03)
LVEF < 50%	117 (52.0)	44 (52.4)	73 (51.8)	0.93	1.03 (0.60–1.76)
AF type					
Paroxysmal	215 (47.6)	83 (45.1)	132 (49.3)	0.39	0.85 (0.58–1.23)
Persistent	36 (8.0)	14 (7.6)	22 (8.2)	0.82	0.92 (0.46–1.85)
Permanent	201 (44.5)	87 (47.3)	114 (42.5)	0.32	1.21 (0.83–1.77)
CKD	181 (40.0)	84 (45.7)	97 (36.2)	0.04	1.48 (1.01–2.17)
eGFR, mL/min/1.73 m^2^	66.9 [52.0–80.5]	66.5 [49.2–80.5]	68.2 [54.2–80.7]	0.35	0.99 (0.98–1.003)
Anemia	72 (15.9)	36 (19.6)	36 (13.4)	0.09	1.57 (0.95–2.60)
Liver failure	16 (3.5)	5 (2.7)	11 (4.1)	0.61	0.65 (0.22–1.91)
CHA_2_DS_2_-VASc score	4 [3,4]	4 [4,5]	3 [3,4]	<0.001	2.61 (2.06–3.32)
CHA_2_DS_2_-VASc score ≥4 points	259 (57.3)	148 (80.4)	111 (41.4)	<0.001	5.82 (3.75–9.01)
CHA_2_DS_2_-VA score	3 [3,4]	4 [3,4]	3 [2,3]	<0.001	4.20 (3.13–5.62)
CHA_2_DS_2_-VA score ≥4 points	173 (38.3)	122 (66.3)	51 (19.0)	<0.001	8.37 (5.44–12.89)
CHADS_2_ score	2 [1,2,3]	2 [1,2,3]	2 [1,2,3]	0.31	0.93 (0.80–1.08)

Notes: CHA2DS2-VASc scale is composed of: C, congestive heart failure (or left ventricular systolic dysfunction); H, hypertension; A2, age 75+ years; D, diabetes mellitus; S2, prior stroke, transient ischemic attack, or thromboembolism; V, vascular disease (e.g., peripheral artery disease, myocardial infarction, aortic plaque); A, age 65–74 years; Sc, Sex category (i.e., female sex). The CHADS2 and CHA_2_DS_2_-VA scales do not take into account V/Sc and Sc, respectively. The vascular disease covers peripheral artery disease, myocardial infarction, or aortic plaque. Abbreviations: AF, atrial fibrillation; BMI, body mass index; CI, confidence interval; CKD, chronic kidney disease; eGFR, estimated glomerular filtration rate; IQR, interquartile range; LVEF, left ventricular ejection fraction; Me, median; n, number; OR, odds ratio; TIA, transient ischemic attack.

**Table 2 jcm-11-03462-t002:** Risk determinants of significant coronary stenosis—direct multivariate logistic regression model.

Variables	OR	95% CI	*p*-Value	OR	95% CI	*p*-Value
		Model 1			Model 2	
Male	1.80	1.14–2.84	0.01	1.76	1.13–2.74	0.01
CHA_2_DS_2_-VA score	3.96	2.96–5.31	<0.001			
CHA_2_DS_2_-VA score ≥ 4 points				7.54	4.87–11.69	<0.001
CKD	1.2	0.76–1.88	0.46	1.32	0.85–2.05	0.22
Anemia	1.33	0.72–2.46	0.36	1.31	0.73–2.37	0.37
Overall prediction rate	75.0%			75.0%		
Sensitivity	66.3%			66.3%		
Specificity	81.0%			81.0%		
Nagelkerke’s R-squared	0.371			0.299		

Abbreviations: CI, confidence interval; CKD, chronic kidney disease; OR, odds ratio.

**Table 3 jcm-11-03462-t003:** AUC for the prediction of significant stenosis.

		95% CI		
Variables	AUC	Lower	Upper	*p*-Value
CHA_2_DS_2_-VA score	0.79	0.75	0.84	<0.001
CHA_2_DS_2_-VA score ≥ 4 points	0.74	0.69	0.79	<0.001
Sex, male	0.59	0.54	0.64	0.001
CKD	0.55	0.49	0.60	0.09
Anemia	0.53	0.48	0.59	0.27

Abbreviations: AUC, area under the ROC curve; CI, confidence interval; CKD, chronic kidney disease; ROC, receiver operator characteristic.

## Data Availability

Data supporting the findings of the study are available from the corresponding author on reasonable request.

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
