# Peer review of "A Newly Defined CHA2DS2-VA Score for Predicting Obstructive Coronary Artery Disease in Patients with Atrial Fibrillation—A Cross-Sectional Study of Older Persons Referred for Elective Coronary Angiography"

_jcm, 2022, doi:10.3390/jcm11123462_

Round 1
Reviewer 1 Report
The idea of this study is quite new - the authors studied the ability to predict obstructive coronary artery disease in older adults with atrial fibrillation. Indeed, new scientific facts were obtained, however, when reviewing the manuscript, a number of questions arose requiring correction and clarification.
Major:
- The authors did not sufficiently study the previous literature on the use of the CHA2DS2-VASc scale in assessing the severity of coronary stenoses. For example, Cetin et al (Prediction of coronary artery disease severity using CHADS2 and CHA2DS2-VASc scores and a newly defined CHA2DS2-VASc-HS score. doi: 10.1016/j.amjcard.2013.11.056) assessed severity using three scales. In another more recent work, Modi et al (CHA2DS2-VASc-HSF score - New predictor of severity of coronary artery disease in 2976 patients. doi: 10.1016/j.ijcard.2016.10.093) also explored this issue. In these works, three groups were distinguished according to the severity of coronary artery lesions (no lesions, insignificant stenoses, significant stenoses), the last group is similar to the group with obstructive lesions studied by the authors. In the article under review, a narrower group of patients (elderly patients with atrial fibrillation) was examined, which allows obtaining new information. However, both in the introduction and in the section Discussion, the authors should compare their data with the above-mentioned works.
- As a rule, when using scales in the study, indicators are given that are included in this scale (for example, doi: 10.1016/j.amjcard.2013.11.056 and doi: 10.1016/j.ijcard.2016.10.093). Authors should include this information in the Methods section.
- In addition to the scale used by the authors (CHA2DS2-VASc), there are other variants of this scale - The CHADS2, CHA2DS2-VASc-HS, and CHA2DS2 -VASc-HSF scores. Judging by the literature data, the last two scales are most correlated with the severity of coronary artery disease. We would like the authors to discuss the possibility of using these variants of the scale in the diagnosis of obstructive lesions in the cohort of studied patients.
- It is a bit strange that in the article on assessing the possibility of detecting obstructive lesions of the coronary arteries, the authors do not at all mention the standard algorithms for examining patients with the assessment of pretest and clinical probability (ECC recommendations of 2013 and 2019 - doi: 10.1093/eurheartj/eht296. and doi: 10.1093/eurheartj/ehz425). First of all, the reader and other researchers would like to understand how the method proposed by the authors for detecting obstructive lesions of the coronary arteries is superior to the traditional examination scheme in the studied cohort of patients.
- I would also like to understand why, with a relatively small predominance of men in the group with stenosis compared with group without stenosis (OR 2.08) at the same time in the multiple logistic regression model, male patients had more than 7-fold greater chance of having significant stenosis. That is, I would like to understand why in this model the risk has increased by more than 3 times? Minor: 1. It seems to me appropriate to link to the information given by the authors in the introduction (lines 83-85). 2. Traditionally, the Discussion section begins with the main results obtained by the authors. I suggest that the authors adhere to such a scheme of discussion. 3. It seems to me that in the discussion the authors evaded the main topic of the work (detection of obstructive lesions of the coronary arteries in patients with suspected coronary heart disease). I propose to revise this section with a more detailed discussion of this issue.
Reviewer 2 Report
Wojszel et al. published a retrospective analysis of AF-pts. receiving a coronary angiography and evaluated the predictive value of CHADS-VASc score to predict significant coronary stenosis. Overall the paper is well written, the graphics have a clear character and the conclusion is suggestive.
A comparison of the predictive value of the CHADS-VASc score to the established Framingham score would be necessary to evaluate the predictive value of CHADS-VASc-Score compared to established scores.
As female sex was associated negatively with the probability of CAD and patients with known CAD were excluded did the authors evaluate the predictive value of "HADS-VA" score?
Furthermore the investigated collective has to be better characterized. How many patients had preinterventional non-invasive diagnostic as modern ESC guidelines implement non-invasive testing before coronary angiography.
Table 2: "Secifitiy" seems like a typo.
Reviewer 3 Report
Wojszel et al enrolled 452 AF patients aged 65 plus years referred to referred to coronary angiography due to exacerbated angina, they found male sex and CHA2DS2-VASc score significantly increased the chance of obstructive CAD, while controlling for chronic kidney disease and anemia. The cut off for CHA2DS2-VASc ≥4 yielded the best combination of sensitivity (80.0%) and specificity (59.0%) for prediction of obstructive CAD in coronary angiography.
The result is interesting and reasonable, because the CHA2DS2-VASc included risk factors for obstructive CAD such as age, hypertension and diabetes mellitus.
I have several questions.
1、The result from this paper suggests that male sex increased the chance of obstructive CAD, however, age is already included in the CHA2DS2-VASc scoring system,how to exclude the interaction between age and CHA2DS2-VASc. Also why not include previous stoke/TIA,Vascular disease history for analyze separately.
2、Did you validate the prediction value of CHA2DS2-VASc ≥4 in the subsequent patients (after 2016) with AF and exacerbated angina who received angiography?
3、How to explain chronic kidney disease and anemia are risk factors for obstructive CAD?
4、what is EFEF in line 292? Is it a typo?
Reviewer 4 Report
In their manuscript, Wojszel & co-workers aimed at determining independent predictors of obstructive CAD in older adults with AF referred for elective CA angiography. Out of 452 patients aged ≥65 years, 184 (41%) presented with significant CA stenosis. Using multivariate regression analysis, the authors found that male sex and CHA2DS2VASc score independently predicted an increased risk of significant CAD.
This is an interesting study. The authors should incorporate CHAD2s risk score as well. This indicator is also relevant on a global scale and should be tested in the univariate and, is feasible, in the multivariate analysis. Given the original approach used by the authors, evaluation of CKD and anemia in addition to CHA2DS2VASc or score, alone or in combination, may add on the predictive value of the risk stratifying scores.
Round 2
Reviewer 1 Report
The authors have done a great job of eliminating my comments and those of other reviewers. As a result, the text of the manuscript has improved significantly, I have no other amendments.
Reviewer 4 Report
The attitude to ignore CHADS2 score in the literature is growing. This reviewer regrets this attitude, as it shows blind acceptance of new rules based on insufficient evidence. The criticism here discloses an even more structural bias by the people who designed the database. Why not to include CHADS2 score? Why not to conceive that a combination made by components of CHADS2 may have more value than CHA2DS2VASc in predicting whatever clinical outcome? This possibility is not even conceived by the authors. Their curiosity about this possibility is cancelled.
To contemplate a database in which the single risk factors are not reported is a significant limit for use of the database for research purposes. In addition, this method does not allow to check for inaccuracy of aggregate data input.
